# Effect of Sampling Method on Detection of the Equine Uterine Microbiome during Estrus

**DOI:** 10.3390/vetsci10110644

**Published:** 2023-11-08

**Authors:** B. A. Heil, M. van Heule, S. K. Thompson, T. A. Kearns, E. L. Oberhaus, G. King, P. Daels, P. Dini, J. L. Sones

**Affiliations:** 1Department of Veterinary Clinical Sciences, School of Veterinary Medicine, Louisiana State University, Baton Rouge, LA 70803, USA; babiche.heil@wsu.edu; 2Department of Veterinary Clinical Sciences, College of Veterinary Medicine, Washington State University, Pullman, WA 99164, USA; 3Department of Population Health and Reproduction (PHR), School of Veterinary Medicine, University of California, Davis, CA 95616, USApdini@ucdavis.edu (P.D.); 4Department of Morphology, Imaging, Orthopedics, Rehabilitation and Nutrition, Faculty of Veterinary Medicine, University of Ghent, 9820 Merelbeke, Belgium; peter.daels@ugent.be; 5Department of Biological Sciences, Louisiana State University, Baton Rouge, LA 70803, USA; thompsonsadiek@gmail.com (S.K.T.); tkearn@lsuhsc.edu (T.A.K.); gkingme@gmail.com (G.K.); 6School of Animal Sciences, Louisiana State University, Baton Rouge, LA 70803, USA; eoberhaus@agcenter.lsu.edu

**Keywords:** microbiome, mare, reproductive tract, sampling techniques, endometrium, uterus

## Abstract

**Simple Summary:**

Bacterial endometritis is among the most common causes of subfertility in mares and has a major economic impact on the equine breeding industry. The sensitivity of detecting microbes using culture-based methods, irrespective of the sample collection method (whether double-guarded swab, low-volume lavage [LVL], or endometrial biopsy) is low, leading to a high rate of false negative samples. Here, using 16S rDNA sequencing, we found that the equine uterus does harbour a distinct microbiome during the estrus phase of the cycle. The microbial community was similar in composition as well as relative abundance at both phyla (Proteobacteria, Firmicutes and Bacteroidota) and genus (*Klebsiella*, *Mycoplasma*, and *Aeromonas*) level, regardless of the sampling technique. The present information about the equine endometrial microbiome could pave the way for innovative treatment methods for endometrial disease and assist sub-fertile mares. This, in turn, could lead to a reduction in the routine use of antibiotics in the equine breeding industry.

**Abstract:**

Bacterial endometritis is among the most common causes of subfertility in mares. It has a major economic impact on the equine breeding industry. The sensitivity of detecting uterine microbes using culture-based methods, irrespective of the sample collection method, double-guarded endometrial swab, endometrial biopsy, or uterine low-volume lavage (LVL), is low. Therefore, equine bacterial endometritis often goes undiagnosed. Sixteen individual mares were enrolled, and an endometrial sample was obtained using each method from all mares. After trimming, quality control and decontamination, 3824 amplicon sequence variants were detected in the dataset. We found using 16S rRNA sequencing that the equine uterus harbors a distinct resident microbiome during estrus. All three sampling methods used yielded similar results in composition as well as relative abundance at phyla (Proteobacteria, Firmicutes, and Bacteroidota) and genus (*Klebsiella*, *Mycoplasma*, and *Aeromonas)* levels. A significant difference was found in alpha diversity (Chao1) between LVL and endometrial biopsy, suggesting that LVL is superior at detecting the low-abundant (rare) taxa. These new data could pave the way for innovative treatment methods for endometrial disease and subfertility in mares. This, in turn, could lead to more judicious antimicrobial use in the equine breeding industry.

## 1. Introduction

Equine endometritis is among the most common causes of subfertility in mares [1]. There are several types of endometritis in the mare, including bacterial, fungal, and persistent post-mating induced [2,3,4,5]. The pathogenesis of bacterial endometritis involves, but is not limited to, colonization of the endometrium by opportunistic bacteria that ascend from the mare’s caudal reproductive tract. The diagnosis of bacterial endometritis is challenging due to the variation in the clinical signs among mares [2]. However, it is crucial to diagnose and treat mares that have bacterial endometritis before mating to ensure pregnancy can be established. Therefore, accurate and timely identification of mares with bacterial endometritis is advantageous to the equine breeding industry in order to avoid increased veterinary cost, overuse of antibiotics, and lost income at yearling sales [1,3].

Endometrial culture and cytology are the most commonly used methods to diagnose bacterial endometritis in the mare. This involves isolation and identification of pathogens and/or inflammatory cells from the endometrium currently utilized by equine practitioners. Several sample collection methods such as a double-guarded endometrial swab, endometrial biopsy, and uterine low-volume lavage (LVL) are used; each has advantages and disadvantages. Culture of a double-guarded endometrial swab is most commonly used in practice for a number of reasons, including the ease of the procedure and minimal processing. However, this technique has a sensitivity (Se, percentage of true positives) of 0.34, and a specificity (Sp, percentage true negatives) of 1.0 [2]. Other diagnostic methods are used less commonly, such as culture and cytology from an endometrial biopsy. This has a higher sensitivity than the double-guarded swab (culture: Se 0.82, Sp 0.92 and cytology: Se 0.77 and Sp 1.0) [2]; however, it is less commonly performed by the equine practitioner due to the labor intensiveness, invasiveness, skills needed, and the requirement of laboratory tissue preparation. It has been suggested that the low sensitivity of both the double-guarded swab and tissue biopsy method is due to the small endometrial surface area sampled, and methods that sample a larger area are superior [6]. The uterine LVL method, where a set volume of sterile fluid is deposited in the lumen of the uterus, massaged and retrieved, samples a larger area of the endometrial surface with an Se and Sp for the culture of 0.71 and 0.86, respectively, and an Se and Sp for cytology of approximately 0.80 and 0.67, respectively [7,8]. Using the techniques mentioned above, the most common bacteria identified in clinical cases of bacterial endometritis are *Streptococcus equi* subsp. *zooepidemicus*, *Escherichia coli*, *Klebsiella pneumoniae*, and *Pseudomonas aeruginosa* [1].

Overall, none of the techniques currently used to diagnose equine endometritis have a high sensitivity; therefore, a high rate of false negatives is expected [2]. The use of traditional diagnostic methods to identify equine bacterial endometritis is further complicated by the fact that >99% of the micro-organisms present in the environment cannot be cultured under standard, aerobic laboratory conditions used in most laboratories [9]. This possibly results in an underestimation of the diversity of the microbes present in the sampled area [10]. Underestimation of microbial diversity has been explored further with the launch of the Human Microbiome Project (HMP) in 2007. In this project, sequence-based techniques were used, and it was found that body sites, which historically were assumed to be sterile, are colonized with their own unique microbiome [10,11]. The techniques used in this project are currently not routinely used to identify bacteria in the mammalian female reproductive tract, but this is an emerging field of research interest [12,13,14,15,16,17,18].

Sequence-based techniques utilize genomic DNA as a cultivation-free method to identify microbes. Genomic DNA can be extracted from swabs, bodily fluids, feces, or biopsies, but it is essential to use sampling approaches that limit the possibility of contamination [19]. The most common sequencing protocol is based on the 16S rRNA gene and hypervariable regions V4–V5, which results in taxonomies accurate at genus and sometimes species levels, and representations of diversity in the form of operational taxonomic units (OTUs) or amplicon sequence variants (ASVs) [14,20,21].

A few studies have been conducted using sequence-based techniques to identify the microbes present in the equine female reproductive tract, and the results have confirmed the presence of a population of anaerobic bacteria, undetectable by conventional aerobic culture, which supports the use of sequence-based techniques to detect a uterine microbiome in the horse [22]. The objectives of this study were to characterize the resident equine uterine microbiome during estrus with 16S rRNA sequencing, using three different, commonly used techniques to diagnose equine bacterial endometritis, including double-guarded endometrial swab, endometrial biopsy, and uterine LVL. We hypothesized that the equine uterus in estrus does harbor a distinct resident microbiome and that a disparity would be present between the different sampling methods due to the area of endometrial surface sampled.

## 2. Materials and Method

### 2.1. Animals

The project was carried out at the School of Veterinary Medicine and the Reproductive Biology Center, Louisiana State University (LSU), Baton Rouge, LA, USA. All horses included in the study were owned by LSU and all procedures were approved by LSU Institutional Animal Care and Use Committee. A total of 16 mixed breed mares aged 13 years (9–18) (median (range)) were included in the study. Prior to and during the project, the mares were housed on pasture and sampled within a 14-day period in July.

#### Inclusion Criteria

To follow common clinical practice, mares in estrus were enrolled in the study based on the following criteria. Estrus was defined as the presence of a follicle >30 mm in diameter, uterine edema, and the absence of corpus luteum detected via transrectal ultrasound, serum progesterone concentration of <1 ng/mL, and an open cervix upon digital palpation. Only mares without signs of endometritis were included. This was determined as <1–2 neutrophils per high-power field cytology brush, no histologic evidence of inflammation or infection (blinded boarded theriogenologist evaluated), no intraluminal uterine fluid present upon transrectal examination during estrus, and a negative aerobic culture of each sample obtained (endometrial biopsy, swab, LVL, and cervical swab).

### 2.2. Methods

After transrectal ultrasound examination, the mares’ perineum was cleaned with 7.5% povidone-iodine scrub (Betadine^®^ Surgical scrub Veterinary, Aviro helath L.P., Stamford, CT, USA) prior to sterile collection of the following samples: external cervical os swab, endometrial swab, endometrial cytology brush, low-volume lavage, and endometrial biopsy, in that order by a single operator. All samples were taken in a clean, climate-controlled, closed examination room.

#### 2.2.1. Sample Collection

##### Endometrial Swab and Cytology Brush

Double-guarded swabs (Minitube, Verona, WI, USA) were used to swab the external cervical os, followed by transcervical introduction of a new double-guarded swab and cytology brush (Minitube). The outer guard of the device remained in the uterus and the cytology brush was removed, followed by the introduction of the swab. A sample was submitted for aerobic culture on MacConkey and blood agar [23] with the Louisiana Animal Disease Diagnostic Laboratory (LADDL), and a duplicate sample was frozen and stored at −80 °C for molecular analyses.

A cytology slide was made with the cytology brush to assess the presence of inflammatory cells using Diff Quick Stain (Heritage Animal Health, Hawarden, IA, USA) [24].

##### Uterine Low-Volume Lavage

Two hundred and fifty milliliters (mLs) Lactated Ringers solution (Baxter Healthcare Corporation, Deerfield, IL, USA) was infused into the uterus through a Foley catheter (Minitube) and transrectally massaged into both uterine horns and the uterine body [7]. Two hundred mLs of the fluid was recovered, a 10 mL aliquot was centrifuged (15,000 G, 15 min), the supernatant was decanted, and the pellet re-suspended in 1 mL PBS, as previously described [25]. An aliquot of the re-suspended pellet was submitted for aerobic culture with the LADDL, and an aliquot was frozen and stored at −80 °C for molecular analyses.

##### Endometrial Biopsy

A sterile biopsy instrument was placed in a sterile rectal sleeve which was perforated in the cervix, as previously described [26]. Two endometrial samples were obtained from the base of each uterine horn. One of the obtained samples was split for culture and cryopreservation at −80 °C for molecular analyses, the second sample was submitted for histology by a blinded, boarded theriogenologist.

##### Negative Control

A sterile, unused swab was submitted for genomic DNA isolation on the same days of sample collection.

### 2.3. DNA Extraction, Sequencing and Metagenomic Analyses

Genomic DNA was extracted from all uterine samples (endometrial swabs, biopsies, and LVL centrifuged pellets) using Qiagen DNeasy PowerSoil extraction kits (Qiagen, Hilden, Germany). Swabs were extracted by removing the swab tips from the applicators with sterile razor blades, and then transferring the swab material directly to bead-beating tubes. Small masses of biopsy samples (approximately 25 mg) were also transferred directly to bead-beating tubes. Lavage fluids were processed by centrifuging the pellet suspended in PBS at 6000× *g* for 10 min at 4 °C. After removing the supernatant, pellets were resuspended in a small volume of bead-beating solution (from Qiagen DNeasy PowerSoil extraction kits) and transferred to bead-beating tubes. Subsequent steps followed the manufacturer’s instructions. In addition to the various uterine samples, a set of blanks was processed similarly as well as a no template control (no sample material was added to the bead-beating tubes). DNA extracts were visualized with gel electrophoresis, transferred to 96-well plates, and then shipped overnight on dry ice to the Research Technology Support Facility of Michigan State University for 16S rRNA sequencing using primers 515f and 806r (V4–V5 region). Barcoding and library preparation were performed and sequencing was carried out on a Miseq platform (Illumina, Inc., San Diego, CA, USA) with 2 × 250 bp paired-end according to the Kozich et al. [27] protocol. All samples were sequenced twice.

Samples were filtered and trimmed based on their quality scores and error rates using the dada2 pipeline [28]. Next, an ASV table was made, and chimeras were removed. The 16S rRNA SILVA v138.1 database [29] was used for mapping and assigning taxonomy. Next, contaminating reads were removed from the samples using Microdecon [30] based on the negative controls (blank and no template control). Downstream analysis was performed using the Phyloseq package, version 1.44.0 [31]. Alpha diversity calculation (Shannon, Chao1, and inverse Simpson), beta diversity (weighted UniFrac), and analysis of similarity (ANOSIM statistic) was performed using the *microbiome*, *amplicon*, *microeco*, and *vegan* packages [32,33,34]. Graphs were generated using *ggplot2*, *dplyr*, *RColorBrewer*, *ggpubr*, and *lattice* packages in R. Bar, and pie plots were generated using Microsoft Excel. Sequences have been deposited in the NCBI SRA as SRP267434.

## 3. Results

### 3.1. Sequencing Results

A total of 16 individual samples were obtained with each sampling method. One endometrial biopsy sample and one uterine LVL sample, from different mares were excluded due to positive aerobic culture. One cervical sample and one endometrial swab sample were not run as the samples were lost. In total, 15 individual samples from each sampling method were sequenced twice. A total of 3968 ASVs from all sample sites were found after quality filtering and mapping. After applying Microdecon, 3824 ASVs were left for downstream analysis.

### 3.2. Alpha Diversity

The microbial communities within the different sample groups were assessed using alpha diversity and compared using ANOVA. No significant difference in alpha diversity between the sampling methods was found for either the Shannon or the inverse Simpson index (*p* > 0.05) (Figure 1). However, the Chao1 index showed a significant difference between the LVL and endometrial biopsy sample, as well as between the LVL and the cervical swab (*p* < 0.05).

### 3.3. Relative Abundance at Phyla and Genus Level

The relative abundance of the most abundant bacteria followed a similar pattern for all sample types at both phylum and genus level (Figure 2, Figure 3 and Figure 4). At phyla level, Proteobacteria, Firmicutes, and Bacteroidota were the most abundant phyla, with a total relative abundance percentage of 82, 83, 80 and 80, respectively, for the cervical swab, endometrial biopsy, endometrial swab, and LVL (Figure 2 and Figure 3). At genus level, the cervical swab, endometrial biopsy, and LVL were dominated by *Klebsiella*, *Mycoplasma*, *Aeromonas*, and *Citrobacter*, while the endometrial swab was dominated by *Klebsiella*, *Mycoplasma,* and *Aeromonas* only (Figure 4 and Figure 5). Further individual differences in the relative abundances of microbes can be seen in all sample groups at both phylum and genus level (Figure 3 and Figure 5). However, with an ANOSIM statistic of 0.1, these individual differences are minimal.

### 3.4. Analysis of Similarity (ANOSIM)

The ANOSIM statistic between all methods was R = 0.1 (*p* < 0.05). The ANOSIM statistic ranges between 0 and 1, and the closer this statistic is to 1, the more dissimilarity is present between the methods (Table 1).

### 3.5. Beta Diversity

The composition of the microbial community between sampling techniques was only significantly different between the cervical swab and endometrial swab (*p* < 0.05, pairwise PERMANOVA on weighted UniFrac distance) (Figure 6). The PERMANOVA tests if there is a significant difference between the sampling methods, and the weighted UniFrac accounts for the abundance of observed organisms and incorporates the phylogenetic distance between microbes in the distance calculation. The weighted UniFrac calculates which fraction of branches on the phylogenetic tree are going to each of the compared groups, but not going to both of them. If the distance calculated is 0, it means that the groups are identical; if the distance calculated is 1, it means that the groups have no taxa in common. All of the groups were overlapping, meaning that all groups shared microbes.

## 4. Discussion

The objective of this study was to characterize the resident equine uterine microbiome during estrus with 16S rRNA sequencing, using three different, commonly used techniques to diagnose equine bacterial endometritis: endometrial double-guarded swab, endometrial biopsy, and uterine LVL. We hypothesized that the equine uterus harbors a distinct resident microbiome in estrus and that a disparity would be present between the different sampling methods.

We revealed that the equine uterus during estrus does harbor a distinct resident microbiome, with Proteobacteria, Firmicutes, and Bacteroidota accounting for 80–83% of the total abundance at the phyla level. This is similar to what was found by Holyoak et al. [35], who found a core microbiome in the equine uterus of cycling mares from different geographical locations. At genus level, the cervical swab, endometrial biopsy, and LVL were dominated by *Klebsiella*, *Mycoplasma*, *Aeromonas*, and *Citrobacter,* accounting for 53–59% of the total abundance at genus level, while for the endometrial swab, 49% of the total abundance was accounted for by *Klebsiella*, *Mycoplasma,* and *Aeromonas.* In disagreement with our hypothesis, the ANOSIM statistic was 0.1, indicating that the microbial composition of the samples is similar between the different sample groups.

The traditionally and most commonly used methods to diagnose bacterial endometritis in the mare, endometrial double-guarded swab, uterine LVL, and culture of an endometrial biopsy, have a variable sensitivity and specificity based on the isolation of the pathogen via traditional culture methods, allowing for false negative results to occur [2]. The incidence of false negatives can further be increased by the presence of dormant bacteria, mainly *S. zooepidemicus*, localized deep within the endometrium [36]. And the presence of biofilm producing bacteria such as *E. coli* and *K. pneumoniae* can also increase the false negative detection rate [2]. In this study, we used metagenetics rather than culture-based techniques; a major difference between these two techniques is that metagenetics, besides live organisms, also identifies dead or fragmented microbial DNA, possibly detecting dormant or biofilm-producing bacteria. Even though dead and fragmented DNA is not replicating in the female reproductive tract, and therefore is not detected by traditional culture methods, it still presents ligands which host cells can recognize; therefore these could contribute to a physiologic interaction with the host [37]. This theory was not further investigated in this current study, and further work is required to investigate the true physiologic interaction, as well as the abundance of dead and fragmented bacterial DNA, with the host. Additionally, >99% of the micro-organisms present in the environment cannot be cultured under standard, aerobic laboratory conditions [9]. Of the most abundant genera present, *Klebsiella*, *Mycoplasma*, *Aeromonas,* and *Citrobacter*, *Citrobacter* and *K. pneumoniae*, are commonly isolated from the mares’ reproductive tract [38]. Furthermore, *K. pneumoniae* is a major pathogen that causes bacterial endometritis in mares [38]. Despite the ability to culture these microbes under standard conditions, they were not identified as culture positive in this study. Furthermore, *Mycoplasma* and *Aeromonas* are not commonly isolated, confirming that standard, aerobic laboratory conditions are likely underestimating the presence of bacteria. In this study, we did not include mares diagnosed with bacterial endometritis. Therefore, we conclude that the bacteria found are present in the healthy equine uterus during estrus. Further studies are required to gain better information about whether bacterial endometritis may be caused by the introduction and establishment of a bacterial population, or if bacterial endometritis could be a result of primary overgrowth or relative overgrowth due to suppression or less succession of other bacterial populations.

Interestingly, despite the large variation in sensitivity and specificity between the sample sites using traditional culture methods, no significant difference was found in the alpha diversity between the different sample types using the Shannon and inverse Simpson index; however the Chao1 index showed a significant difference between LVL and endometrial biopsy sample and between LVL and cervical swab. The inverse Simpson index is a dominance index which gives more weight to common or dominant species; as a result, the presence of a few rare species will not affect the diversity found. The Chao1 index, however, puts more weight on singletons and rare ASVs; therefore, the presence of rare species will also result in a higher diversity index. Thus, as the Chao1 index, but not the Shannon and inverse Simpson indices, was significantly different, this difference is likely because of the low-abundant taxa found in using the LVL sampling method. Because the Chao1 index was higher in the LVL samples, it suggests that LVL is better at finding low-abundant taxa and rarer ASVs than the endometrial biopsy. A reason for this could be that the endometrial biopsy sample contains a high number of eukaryotic host reads from the equine endometrium; these host reads could make it more difficult to isolate rare bacterial DNA reads present in the sample. The cervical swab sample was a control to assess to what extent the external cervical os shares a microbial community with the endometrium. Therefore, it is not surprising that the cervical sample was significantly different from the other samples.

The relative abundance of the most abundant bacteria follows a similar pattern for all sample types in this study, with Proteobacteria, Firmicutes, and Bacteroidota accounting for 80–83% of the total abundance at phyla level. This is similar to the findings of Holyoak et al. [35], who found a core microbiome in the equine uterus of mares from different geographical locations consisting of Proteobacteria (~48%), Firmicutes (30%), Bacteroidetes (12%), Actinobacteria (5%), Tenericutes (2%), and Kiritimatiellaeota (1%). Similar profiles have been found in women, where Proteobacteria, Firmicutes, Actinobacteria, and Bacteroidetes have been found to dominate the endometrial microbiome [38]. At genus level, the cervical swab, endometrial biopsy, and uterine LVL were dominated by *Klebsiella*, *Mycoplasma*, *Aeromonas*, and *Citrobacter,* accounting for 53–59% of the total abundance at genus level, while for the endometrial swab, 49% of the total abundance was accounted for by *Klebsiella*, *Mycoplasma,* and *Aeromonas.* This contrasts with the dominant genera found in the human endometrium, which has an abundance of *Lactobacillus, Gardnerella*, *Prevotella*, *Atopobium*, and *Sneathia* [11]. At phyla level, the three top phyla account for 80–83% of the total abundance, and the remainder is made up of many different phyla. At genus level, the top four genus account for 53–59% of the total abundance, depending on the sample method. This is reflected in the significance of *p* < 0.05 for the ANOSIM statistic with R = 0.1, indicating that the compared groups are similar in composition; however, a statistically significant difference is present in the microbial community presented by less abundant taxa.

The similarities found between endometrial samples (swab, LVL, and biopsy) and the cervical samples are not surprising since during estrus, the cervix is open, allowing free communication with the uterus. The mare is known to have a core vaginal microbiome, consisting of Firmicutes, Bacteroidetes, Proteobacteria, and Actinobacteria, and at genus level, this vaginal microbiome was defined by *Porphyromonas*, *Campylobacter*, *Arcanobacterium*, *Corynebacterium*, *Streptococcus*, *Fusobacterium*, uncultured *Kiritimatiaellae,* and *Akkermansia* [39]. Human studies have shown that the genital tract microbiome progressively changes from vagina to endometrium and that a strong microbiome correlation is present between vaginal, cervical, and endometrial samples within individuals [38]. The outer cervical os samples in this study had a similar core microbiome at phyla level to the vaginal microbiome found by Barba et al; however, at genus level, the outer cervical os was closer to the endometrial microbiome than the vaginal microbiome. This suggests a similar progressive change from vagina to endometrium may be present in the equine microbiome detected using metagenetics to that observed in human studies.

However, beta diversity, calculated using weighted UniFrac, showed a significant difference between the cervical and endometrial swabs (*p* = 0.01). This may be explained due to the higher abundance of phyla and genera found in the cervical sample compared to the endometrial sample, and, similar to in women, a progressive change in microbiome between the open cervix and endometrium might be present. Furthermore, the possibility of cross-contamination from the cervix in the other samples could be present, causing them to have a more similar beta diversity. The endometrial swab was the first sample taken after sampling the external cervical os. On consecutive samples, instruments had already been passed through the cervix, causing a possible small amount of carry-over of microbes from the cervix to the endometrium and sampling instruments. However, this possibility exists for all transcervical procedures and warrants further examination.

In humans, a consistent association has been observed between dysbiosis of the vaginal microbiome and unfavorable reproductive outcomes. The similar microbial richness seen between the external cervical os and the uterus in both women and mares proves that further studies into both endometrial and vaginal and cervical microbiomes in mares are warranted [14]. In women, it is known that the composition of the vaginal microbiome is highly dynamic and influenced by differences in estrogen levels, puberty, menstruation, and sexual activity [10,40]. Therefore, further research is needed to determine the influence of reproductive hormones on the equine reproductive tract microbiome and subsequent reproductive success or failure.

## 5. Conclusions

In conclusion, as determined using 16S rRNA sequencing, the equine uterus does harbor a distinct resident microbiome during estrus, which for all three sampling methods used (endometrial swab, endometrial biopsy, and LVL), is similar in both composition and relative abundance at both phyla (Proteobacteria, Firmicutes, and Bacteroidota) and genus (*Klebsiella*, *Mycoplasma*, and *Aeromonas)* level.

There was a significant difference in alpha diversity (Chao1) between uterine LVL and endometrial biopsy and between LVL and cervical swab, suggesting that LVL is better at sampling low-abundant (rare) taxa. It is not clear if these taxa are relevant in clinical practice. There was also a significant difference in beta diversity between endometrial swab and cervical swab, but not between the other sampling methods. This result may be due to transient carry-over of microbial DNA with repeated sampling. More research is needed to correlate these results with diagnosis and clinical management of endometritis in the mare. Our results suggest that all the tested methods can be used for downstream analysis of endometrial DNA composition. However, LVL samples are better for detecting low-abundant taxa.

## Figures and Tables

**Figure 1 vetsci-10-00644-f001:**
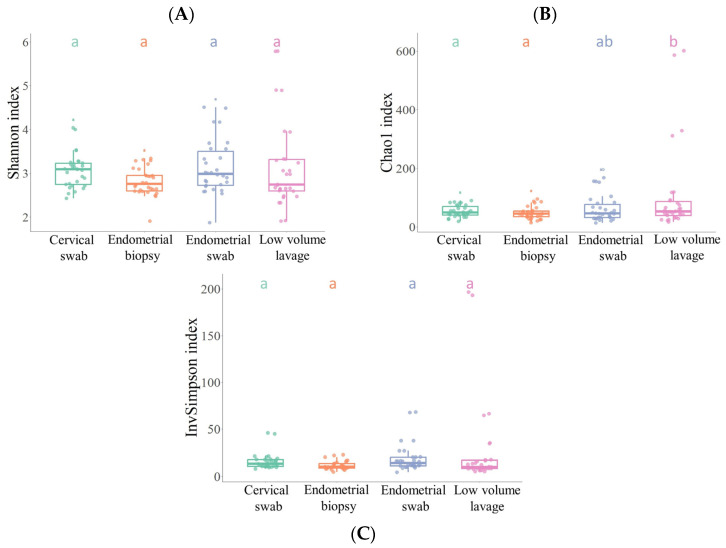
Alpha diversity index in the four endometrial and cervical sample groups. (**A**) Shannon index, (**B**) inverse Simpson index and (**C**) Chao1 index between cervical swab, endometrial biopsy, endometrial swab, and low-volume lavage. Differing superscripts (a,b) within box plots are different (*p* < 0.05).

**Figure 2 vetsci-10-00644-f002:**
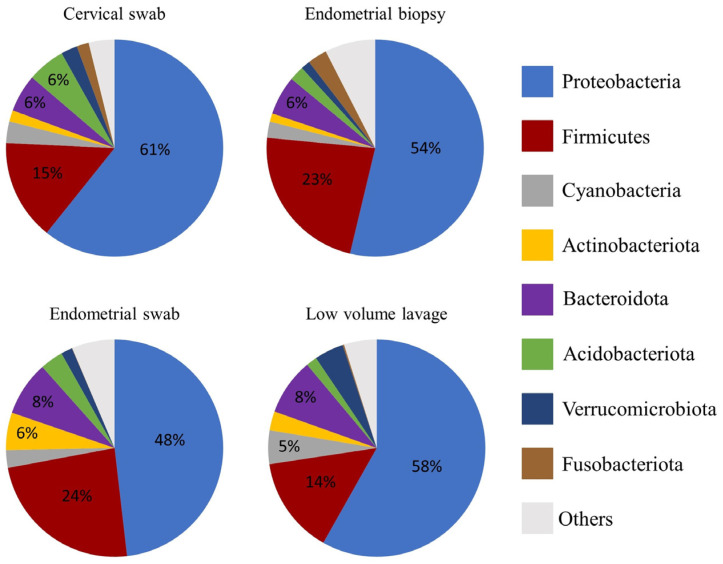
Relative abundance on phylum level in the four sample groups. Proteobacteria, Firmicutes and Bacteroidota were found to be the three most abundant phyla in all samples. Other abundant phyla are Acidobaceriota and Cyanobacteria.

**Figure 3 vetsci-10-00644-f003:**
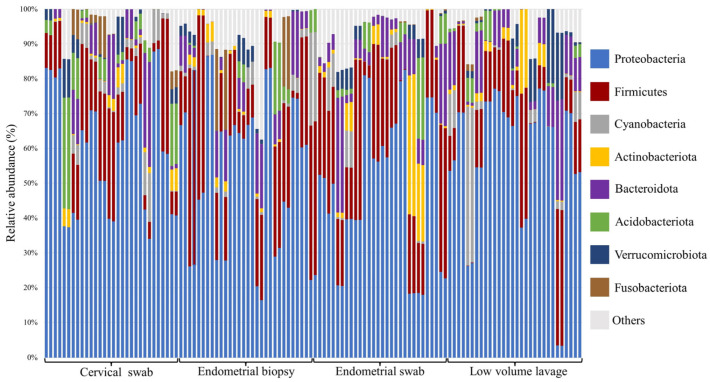
Relative abundance on the phylum level in the four sample groups. Proteobacteria, Firmicutes, Bacteroidota, Acidobaceriota, and Cyanobacteria are the most abundant phyla. There are individual differences between samples in all sample groups.

**Figure 4 vetsci-10-00644-f004:**
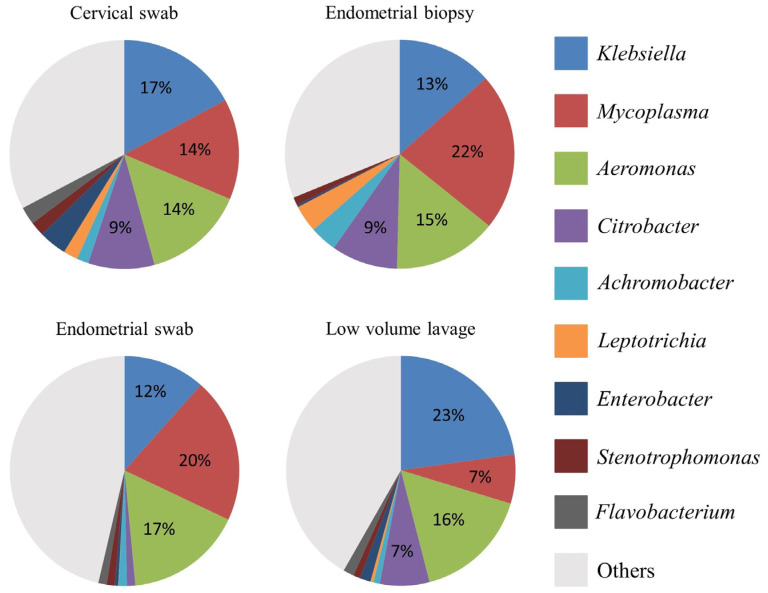
Relative abundance on genus level in the four sample groups. The most abundant genera in all samples are *Klebsiella*, *Mycoplasma*, *Aeromonas*, and *Citrobacter*.

**Figure 5 vetsci-10-00644-f005:**
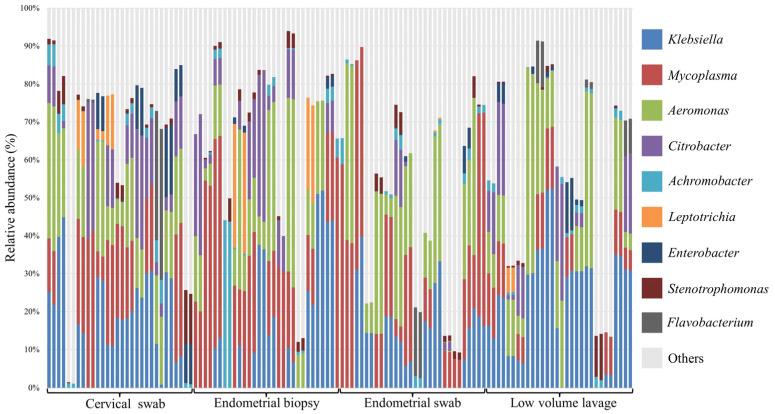
Relative abundance on of top 200 genera in the four sample groups. The most abundant genera in all samples are *Klebsiella*, *Mycoplasma*, *Aeromonas*, and *Citrobacter*. There are individual differences between samples in all sample groups.

**Figure 6 vetsci-10-00644-f006:**
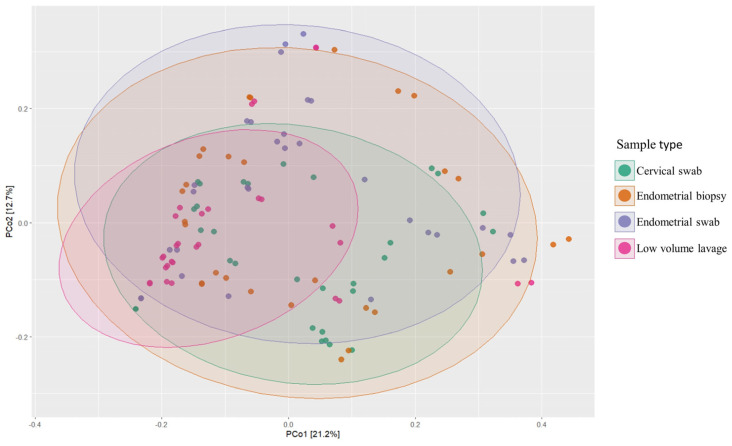
PCoA plot of weighted UniFrac distance. A Significant difference in beta diversity was found between cervical swab and endometrial swab (*p* < 0.05).

**Table 1 vetsci-10-00644-t001:** Comparison of sample methods and their *p*-values.

Sample Method 1	Sample Method 2	*p*-Value
Endometrial biopsy	Endometrial swab	0.016
Endometrial biopsy	LVL	0.001
Endometrial swab	LVL	0.001

## Data Availability

Sequences have been deposited in the NCBI SRA as SRP267434.

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
