# Peer review of "Effect of Sampling Method on Detection of the Equine Uterine Microbiome during Estrus"

_vetsci, 2023, doi:10.3390/vetsci10110644_

Round 1
Reviewer 1 Report
Comments and Suggestions for Authors
General overview: The study aim of Heil et al. was to characterize the resident equine uterine microbiome during estrus, with 16S rRNA-sequencing using three commonly used techniques to diagnose equine bacterial endometritis, including double-guarded endometrial swab, endometrial biopsy, and uterine LVL, suggesting that all the tested methods can be used for downstream analysis of endometrial DNA composition.
The results are coherently presented from a scientific point of view, accompanied by a careful study of the literature in the introduction section and motivated in the discussion and conclusion sections. I have only a few minor remarks and suggestions that I hope the authors will appreciate to improve an already satisfying work.
Minor Remarks
Materials and Methods section:
Pay attention to the titles of the subparagraphs: I advise you to choose whether to write them all in italics or not (e.g. subparagraphs 2.1.1., etc etc)
Figure 1: I advise authors to enlarge the font of the figure, also trying to create a larger panel, wherever possible
Figure 2,3,4,5,6: I advise authors to enlarge the font of the figures; pay attention to the figure legends: is there a formatting problem? The characters are not readable well.
Table 1: I advise authors to write the “p-value” in italics.
Author Response
On behalf of all the co-authors, I'd like to thank you for the thoughtful comments. We have addressed them all including formatting suggestions and p values in the table.
Reviewer 2 Report
Comments and Suggestions for Authors
This is an interesting study and provides important information in the accuracy of different techniques of sampling the uterine microbiome. Please find my general and specific comments below.
General comments:
Your main goal was to characterize the efficacy of the different methods of sampling the uterine microbiome in detecting the different bacteria, with hopes of it being used to diagnose endometritis. Although you did not have a group with endometritis to compare the efficacy of those methods to detect the bacteria associated with endometritis, what insights can be taken from the bacteria found in your samples to endometritis? Are the bacteria responsible for endometritis present in the healthy microbiome, and disruptions in the microbiome is what allows for those bacteria to thrive and cause disease, or are these bacteria simply not present at all and must be introduced in the uterus?
Also, I would argue that using LVL allows you to acquire information related to the function of the microbiome (i.e. environmental conditions). While using the 16S rRNa technique allows you to detect which individuals are present, using LVL allows you to collect data that belong to the environmental conditions (i.e. pH, ROS, cytokine concentrations) which will essentially determine which bacteria will thrive, and could potentially be included in the diagnostic procedure. (For example in humans, a low vaginal pH is considered healthy because it indicates great abundance of lactobacilli, which is known to keep the acidic conditions in the vagina, a condition that lactobacilli bacteria will thrive but pathogenic bacteria will not, and thus protect the host from infections).
Specific comments:
Materials and Methods:
Lines 102-103: Were all mares kept in the same pasture?
Discussion:
Lines 248-261: These fragmented/dead bacteria are not replicating. Wouldn't you expect these bacteria to be in lower abundance? If so, did you detect them in lower abundance?
Also, what was the abundance of these pathogenic "dormant bacteria" in the samples from your healthy mares? Are these bacteria present even in healthy individuals, and an immunosuppression allows them to replicate and cause the disease, or are they only present during the disease?
Comments on the Quality of English Language
Line 294: Rephrase "This is similar was found"
Line 298: Avoid starting the sentence with "And".
Author Response
On behalf of all the co-authors, I'd like to thank the reviewer for their thoughtful comments and suggestions. Your general ones have been incorporated into the Discussion. Your remarks on the LVL are thought provoking and we are excited to follow up on that in a subsequent manuscript. Your specific comments have been edited accordingly in the Methods and Discussion.